# Identification, Characteristics and Function of Phosphoglucomutase (PGM) in the Agar Biosynthesis and Carbon Flux in the Agarophyte *Gracilariopsis lemaneiformis* (Rhodophyta)

**DOI:** 10.3390/md20070442

**Published:** 2022-07-02

**Authors:** Qionglin Chen, Xinlei Yu, Shixia Liu, Suya Luo, Xiaojiao Chen, Nianjun Xu, Xue Sun

**Affiliations:** Key Laboratory of Marine Biotechnology of Zhejiang Province, School of Marine Sciences, Ningbo University, Ningbo 315211, China; chenqionglin2333@163.com (Q.C.); yuxinlei19942021@163.com (X.Y.); a986644857@163.com (S.L.); chenxiaojiao@nbu.edu.cn (X.C.); xunianjun@nbu.edu.cn (N.X.)

**Keywords:** agar biosynthesis, carbon flux, floridean starch, (iso)floridoside, *Gracilariopsis lemaneiformis*, phosphoglucomutase (PGM)

## Abstract

Agar is widely applied across the food, pharmaceutical and biotechnology industries, owing to its various bioactive functions. To better understand the agar biosynthesis in commercial seaweed *Gracilariopsis lemaneiformis*, the activities of four enzymes participating in the agar biosynthesis were detected, and phosphoglucomutase (PGM) was confirmed as highly correlated with agar accumulation. Three genes of *PGM* (*GlPGM1*, *GlPGM2* and *GlPGM3*) were identified from the *G. lemaneiformis* genome. The subcellular localization analysis validated that GlPGM1 was located in the chloroplast and GlPGM3 was not significantly distributed in the organelles. Both the GlPGM1 and GlPGM3 protein levels showed a remarkable consistency with the agar variations, and GlPGM3 may participate in the carbon flux between (iso)floridoside, floridean starch and agar synthesis. After treatment with the PGM inhibitor, the agar and floridean starch contents and the activities of floridean starch synthase were significantly decreased; products identified in the Calvin cycle, the pentose phosphate pathway, the Embden-Meyerhof-Parnas pathway and the tricarboxylic acid cycle were depressed; however, lipids, phenolic acids and the intermediate metabolites, fructose-1,6-phosphate were upregulated. These findings reveal the essential role of PGM in regulating the carbon flux between agar and other carbohydrates in *G. lemaneiformis*, providing a guide for the artificial regulation of agar accumulation.

## 1. Introduction

Agar, as a major important component of the extracellular matrix and the cell walls in some red algae, has great importance in the adaptation to marine environment. In addition, this sulfated polysaccharide possesses various bioactive functions, which are now widely applied across the food, pharmaceutical and biotechnology industries [1,2]. Approximately 60–80% of agar is contributed from the seaweeds *Gracilaria/Gracilariopsis*, which are the main genus of agar production. Among these agarophytes, *Gracilariopsis lemaneiformis* has a good agar yield of 10–30% and a gel strength of 140–1600 g·cm^−2^ [3,4]. Although the agar from *G. lemaneiformis* has a superior yield and quality, there are still many problems to be solved, such as the complex agar extraction method, the high demands for raw materials and the incomplete agar accumulation pathway; all of these problems hinder the development of the agar industry.

In the process of agar biosynthesis, two galactoses of β-D-galactose and 3,6-anhydro-L-galactose are first synthesized by a series of enzymes [5]. Then, the liner backbone of agar is cross-linked, using these two galactoses through β-1,3- and α-1,4-glycosidic bonds by galactosyltransferases (GTs), respectively [3,6]. In red algae, many studies have proposed the relationship between agar content and the transcriptional expressions of its synthetic enzymes. The gene expressions of galactose-1-phosphate uridylyltransferase (*GALT*) and UDP-glucose pyrophosphorylase (*UGPase*) are positively correlated with the agar content of *G. lemaneiformis* [7,8]. The transcriptional levels of GDP-mannose-3’,5’-epimerase (*GME*) in *Gracilaria changii* and *Gracilaria salicornia* exhibit different expression modes, corresponding to the different agar contents [9]. In *G. lemaneiformis*, the transcriptional expression level of *PGM* is proved to be highly related to agar biosynthesis [10]. However, few studies focus on the correlation between the enzyme activities and the agar content, as the enzymes play important and direct functions in agar biosynthesis.

Agar is situated at the end of carbon metabolism, does not decompose, but other metabolites, such as floridean starch and floridoside, can decompose into glucose-1-phosphate (G1P) and glucose-6-phosphate (G6P) that may be reused in the next round of synthesis of other carbon metabolites. Phosphoglucomutase (PGM), belonging to the phosphohexose mutase family, can regulate and control the convention of G1P and G6P, and be regarded as a major branch point of carbon metabolism in the formation of storage polysaccharides and energy production. In most higher plants, PGM plays a main role in the starch and sucrose synthesis [11]. However, (iso)floridoside is the main carbon storage material in Rhodophycea and plays a role, as does sucrose in higher plants. Furthermore, red algae constitute an exception to the rule, because their biosynthesis of starch takes place in the plastid, unlike chlorophytes, who synthesizes and stores starch as granules outside their plastids in the cytosol [12]. Due to the different carbon assimilation mechanisms, the role of PGM in carbon flow in red algae is still obscure. 

In recent years, the important economic macroalga *G. lemaneiformis* has expanded rapidly along the coastal regions of China, which is mainly used for agar production, abalone bait and food industries. In addition, *G. lemaneiformis* has a high ability to remove and sequester carbon to attain carbon neutrality, which demonstrates the substantial ability of this seaweed in carbon use [13]. In this study, we screened one enzyme (PGM) connected with agar biosynthesis, characterized the *GlPGM*s, verified the enzyme’s subcellular location and expression modes, and analyzed the carbon flux among the diverse carbon storage materials under PGM-inhibited conditions in *G. lemaneiformis*. This study has an important guiding role in the artificial regulation of algal agar accumulation.

## 2. Results

### 2.1. Correlation Analysis between Enzyme Activities and Agar Contents

The activities of PGM, UGPase, mannose-6-phosphate isomerase (PMI), phosphomannomutase (PMM) and agar contents were simultaneously measured and compared in the different conditions (Appendix A). Of the four enzymes, PGM had the highest Pearson correlation coefficient, with 0.973 and 0.999 in different cultivation months and fields, respectively (*p* < 0.05). In the nitrogen treatment, PMI, PGM and PMM showed the significant correlations with 0.896, 0.955, 0.888 on day 7, respectively. UGPase showed the highest positive correlation (0.864) with agar content on day 14; PGM revealed the positive correlation with agar content during the nitrogen treatment period. In the salinity treatment, only PGM showed a positive correlation at all of the time points. Combining with the results of the four enzyme activities and the agar content of *G. lemaneiformis* in different cultivation conditions, the PGM activities had the highest positive correlation (Table 1). 

### 2.2. Bioinformatic Characteristics of GlPGM

The ORFs of *GlPGM1*, *GlPGM2* and *GlPGM3* encode 585 aa, 580 aa and 572 aa, respectively. The molecular weights of GlPGM1, GlPGM2 and GlPGM3 are 63.14 kDa, 62.68 kDa and 61.97 kDa, with a theoretical isoelectric point (pI) of 5.02, 5.05 and 6.33, respectively. The conserved PGM_PMM_I domains with 141 aa in GlPGM1 and 146 aa in GlPGM2 are located at the N-terminal of the two GlPGMs. The conserved PGM_PMM_II domain is at the C-terminal of GlPGM3. 

A phylogenetic PGM tree and conserved motifs were constructed to show the evolutionary relationship between the related PGM sequences from large classes of plants (Figure 1). The tree is separated into three main clades, according to the conserved motifs. GlPGM1, GlPGM2 are located in the clade I, which are close to red algae, followed by brown algae, green algae and higher plants; GlPGM3 is located in the clade III and is close to red algae, followed by higher plants. The tree and motifs indicate that these PGM sequences in the same clade have similar structural characteristics.

### 2.3. Subcellular Location of GlPGM1 and GlPGM3

The online prediction tools (CELLO, WoLF PSORT and Plant-mPLoc) agreed that GlPGM1 and GlPGM2 were located in the chloroplast. For GlPGM3, the CELLO Prediction showed that it was in the cytoplasm with 0.802 reliability, while WoLF PSORT revealed that it was located in the mitochondrion and Plant-mPLoc predicted it was in the chloroplast.

Immunocolloidal gold transmission electron microscopy validated the locations of GlPGM1 and GlPGM3 in *G. lemaneiformis* (Figure 2). To better observe the subcellular localization of PGMs, we firstly performed a structural observation of *G. lemaneiformis* without incubation by antibodies (Appendix A). The chloroplast surrounds the inside of the cell and the floridean starch is found in the cytoplasm, surrounding the nucleus. According to the quantitation of the antigen locations and density, more of the colloidal gold particles were distributed in the chloroplast (9.33 ± 0.54/μm^2^) than in the cytoplasm (0.50 ± 0.06/μm^2^) (*p* < 0.05). However, the distributions of the colloidal gold particles of GlPGM3 were not significantly different between the cytoplasm, chloroplast or mitochondrion.

### 2.4. Immunoblotting Analysis of GlPGM1 and GlPGM3

In the salinity and nitrogen treatments, the expressions of GlPGM1 in low salinity and nitrogen-limited treatments were significantly higher than that in salinity 33 and normal nitrogen treatments, respectively. For different cultivation fields, the activities of GlPGM1 in *G. lemaneiformis* in Xiangshan (XS) were higher than that in other fields. The activities of GlPGM1 were the highest in the summer (May and July), and the lowest in the winter (January and March). The expressions of GlPGM3 in XS were significantly higher than that in other fields (*p* < 0.05). The GlPGM3 was significantly depressed in September and November (Figure 3).

### 2.5. Changes of Enzymic Activities under PGM Inhibited Condition

Treated with Disperse Blue 56, a parabolic and noncompetitive inhibitor of PGM [14], the enzymic activities and agar content were significantly decreased by approximately 40% at 5 μM of inhibitor concentration during the 24 h period (Appendix A). Thus, the inhibitor concentration of 5 μM was used for the further experiments. 

During the 24 h period, the activities of UGPase showed no significant variations in the control and inhibitor treatment. Similarly, the changes in the activities of the trehalose-6-phosphate synthase (TPS) and the α-galactosidase (GLA) of (iso)floridoside were not significantly different from the above enzymes, the activities of the floridean starch synthase (SS) in the control were significantly higher than that in the inhibitor treatment at 24 h, and the activities of PMI were increased significantly at 3 h but repressed at 6 h (Figure 4).

### 2.6. Changes of Primary Carbon Metabolism under PGM Inhibited Condition

A total of 503 metabolites in the two treatments were identified, including 147 lipids, 76 organic acids, 60 nucleotides and derivatives, 70 phenolic acids, 97 amino acids and derivatives and 52 other compounds. The principal component analysis (PCA) was subsequently used for the multivariate statistical analysis. As shown in Figure 5A, the samples of the two treatments were separated in the PCA plots. The two principal components interpreted 54.79% of the total variance, of which the first principal component (PC1) mainly separated the groups under the control and inhibited treatments, with 30.45% variance contribution value. The heatmap analysis showed the accumulation pattern of the different classes of metabolites in *G. lemaneiformis* in different treatments (Figure 5B). The result revealed that many of the metabolites were downregulated under the PGM-inhibited conditions. However, the PGM inhibitor upregulated the other materials, especially lipids and phenolic acids. 

To further understand the role of PGM during the agar accumulation, the changes of the metabolites in the tricarboxylic acid cycle (TCA) cycle, the pentose phosphate pathway (PPP), the Embden–Meyerhof–Parnas pathway (EMP), the Calvin cycle (CC) and the agar pathways were compared (Figure 6). The PGM inhibitor downregulated the metabolites related to the agar biosynthesis and carbon metabolic pathways. The contents of (iso)floridoside, floridean starch and agar were depressed, with fold changes of 0.85, 0.65 and 0.61, respectively. However, fructose-1,6-phosphate in the EMP pathway were significantly upregulated, with a fold change of 1.74. 

## 3. Discussion

### 3.1. The Enzymes in Agar Biosynthesis Pathway

The yield of agar can be altered with the changing environment. Low salinity, nitrogen deficiency and phosphate enrichment can significantly promote the agar contents [8,15]. Here, the agar accumulated after low salinity and nitrogen-deficient treatments. Numerous reports show the effects of the season on agar yield and the properties of *Gracilaria*; however, the results are contradictory, partly because of the different methods used for agar extraction, and the nature of seasonal changes varies geographically. The lowest and highest yields of *Gracilaria veleroae* agar occurred in the summer and autumn, respectively, while the highest agar yields for *Gracilaria vermiculophylla* occurred in the winter [16]. A significant increase in the agar content in *G. lemaneiformis* was found in summer (May and July in 2020) and a richer agar content occurred in Xiangshan. 

The enzymes in agar synthesis can modulate the content of substrates and thus affect the agar content. The four enzymes measured in this study showed positive correlations with agar variations under different treatments, and PGM exhibited the significant correlation with agar content. PGM may regulate the carbon flux between the different materials in marine algae. In *G. lemaneiformis*, the expression of *PGM* is closely related with the agar content after application of 24-epibrassinolide [11], which is consistent with our studies. UGPase is another enzyme which is associated with the agar content. The positive correlations between the agar content and the transcriptional level of *UGPase* could also be found at different life stages and in different cultivation fields [8,17], which is a candidate molecular marker for evaluating agar content. PMI and PMM are involved in the GDP–mannose pathway, another pathway of agar synthesis. The negative relevance between their activities and agar synthesis was found under the salinity and nitrogen treatments in *G. lemaneiformis*. These variations of relevance may be because other materials share the same upstream of the pathway and agar is at the end of metabolism.

### 3.2. Characteristics and Functions of GlPGM

With the development of high throughput sequencing and bioinformatics technology, a large number of gene families have been excavated. Numerous *PGM* genes are screened in *Arabidopsis thaliana*, *Oryza sativa* and *Zea mays*. The numbers of *PGM* genes in seaweeds are relatively less than those in higher plants. Four PGM sequences are found in red algae, such as *Gracilariopsis chorda*, *Chondrus crispus* and *Porphyra umbilicalis*. However, no PGM gene is found in brown seaweed, while the bifunctional enzyme PMM can function as PMM and PGM [18]. A total of three *PGM* sequences were obtained through screening the genome database of *G. lemaneiformis.* In this study, the whole evolutionary tree of the PGM family is divided into three clades, because of the conserved motif. There is a close sequence similarity between these PGMs in different species and they are descended from ancestral gene via gene duplication and speciation events. GlPGM1 and GlPGM2 have a similar conserved motif and are clustered into the clade I, from which it can be concluded that the homologous PGMs are similar in the biological role. GlPGM3 is close to red algae, followed by higher plants, which indicates that these PGM sequences from higher land plants could be evolved from red algae. Meanwhile, the tree implies that gene duplication can give rise to the production of in-paralogs in a species and also out-paralogs in different species [19]. 

Two isoforms, plastidial and cytosolic PGMs, display high activities in potato tubers, *Arabidopsis* [20], tobacco leaves [21] and *Zea mays*. The plastidial PGM in potatoes forms part of the starch in the generated transgenic potato plants [22]. The plants with reduced cytosolic PGM activity are characterized by a dramatic reduction in tuber sucrose and even more complex changes on a range of diverse metabolic pathways [23]. However, PGM functions are still obscure in marine algae, as the polysaccharides in higher plants are different from seaweeds and the unique starch and (iso)floridoside metabolisms in Rhodophycea. GlPGM1 is observed located in the chloroplast, which is consistent with the results from the online predictions. The floridean starch was found in the cytoplasm, surrounding by the nucleus, which is different from the starch observed in the chloroplast in higher plants. The extra-plastidic starch synthesis in this alga proceeds via a UDP glucose-selective α-glucan synthase, in analogy with the cytosolic pathway of glycogen synthesis in other eukaryotes [12,24]. However, the similar expressional patterns of GlPGM1 with agar variations were found under different conditions. Triose-phosphate is reported to be produced by the Calvin cycle in plastids and be transported to cytosol for (iso)floridoside, floridean starch and agar synthesis [25]. The role of GlPGM1 in carbon assimilation or whether it is related to agar, floridean starch and (iso)floridoside synthesis still need to be further explored. The localization of GlPGM3 in each organelle did not differ significantly, which is consistent with the predictions online. The Western blot analysis of GlPGM3 validated the same trend to agar variation, except for the changes in the different months. According to the biosynthesis pathways of agar, floridean starch and (iso)floridoside in red algae, they share the same upstream enzymes in the pathway, including glucose-6-phosphate isomerase (PGI), PGM and UGPase [26], (iso)floridoside can be decomposed to provide the precursor for floridean starch and agar synthesis. On the other hand, these three carbohydrates compete the same synthetic substrates, such as UDP-glucose and galactose-1-P [27]. The inconsistency between the expression patterns of GlPGM3 and agar changes may be related to the carbon flow between these three carbon storage substances.

### 3.3. Agar and Carbon Flux Affected by PGM Inhibitor

To better explore the role of PGM in the pathways of agar biosynthesis and carbon flux, different inhibitors of PGM were chosen. Some of the endogenous intermediates and related structural analogs, such as adenosine triphosphate (ATP), citrate [28], fructose 2,6-bisphosphate and xylose 1-phosphate [29,30], and some trace metals, such as vanadate [31], are all proposed to have inhibitory effects on PGM. Disperse Blue 56 is identified by structure-based screening methods, and can solely inhibit the activity of PGM by aggregation [14]. Thus, in this study, we chose Disperse Blue 56 as a PGM inhibitor, and changes in the primary metabolites and the activities of enzymes related to agar pathway were further detected.

SS is a key regulatory enzyme involved in the starch biosynthesis and is differently regulated in different tissues [32]. The activities of SS and the starch content in this study were significantly decreased, owing to the PGM inhibitor. As enzyme activities and pathway flux are commonly related and PGM is essential for starch synthesis, it provides evidence for our conclusion that PGM catalyzes the sequential reactions in the biosynthesis of starch in *G. lemaneiformis*. In addition, the activities of PMI varied due to the PGM inhibition. However, other enzymic activities and related metabolites were not detected, so the function of PGM on 3,6-anhydro-L-galactose synthesis should be further analyzed. 

In the present study, the inhibition of PGM resulted in the overall increase in the lipid contents while a decrease in carbohydrates, such as (iso)floridoside, floridean starch and agar, which implied a possible carbon flux between the lipids and the energy storage materials in *G. lemaneiformis*. Earlier studies have shown that carbohydrates and the lipid biosynthetic pathways compete for the substrates, as the biosynthetic pathways utilize the same precursors. PGM overexpression can accumulate the production of chrysolaminarin, while the lipid contents are reduced [21]. The carbon partitioning from starch to lipid synthesis is also found for long-term energy storage [33]. In *G. lemaneiformis*, (iso)floridoside, starch and agar accumulations are the main carbon sinks in the cells. It is also important to know how carbon flow is regulated by the carbohydrate and lipid metabolism in Rhodophyta, which is important for biofuel manipulations [24]. 

For the carbon flux under PGM inhibited condition, the Cavin cycle, the EMP pathway, the TCA cycle and agar biosynthesis showed a downward trend, whereas fructose-1,6-phosphate exhibited a significant increase. During the biological pathway, fructose-1, 6-phosphate is an important intermediate in EMP and PPP. It can be cleaved into two trioses, dihydroxyacetone phosphate and glyceraldehyde-3-phosphate, which are the substrates of carbohydrates and lipid biosynthesis [34]. In the agar biosynthetic pathway, glyceraldehyde-3-phosphate can be converted to fructose-1,6-bisphosphate by using fructose-1,6-bisphosphate aldolase, and then fructose-6-phosphate, which is a central metabolite used for the synthesis of agar monomer, and is generated by fructose-1,6-bisphosphatase [35]. Fructose-1,6-bisphosphatase is a rate-limiting enzyme in gluconeogenesis, catalyzing the irreversible conversion of the substrate fructose-1,6-bisphosphate into fructose-6-phosphate. It is also involved in many of the other metabolic pathways, such as the Calvin cycle and starch biosynthesis [36]. Combining all of these analyses, we speculate that PGM inhibition leads to an overall decrease in respiratory metabolic pathways, in which fructose-1,6-bisphosphate accumulates due to the restriction of fructose-1,6-bisphosphatase, resulting in a decrease in the downstream carbohydrates, such as floridean starch and agar contents and an increase in the lipids.

## 4. Materials and Methods

### 4.1. Algal Materials and Culture Condition

The alga of *G. lemaneiformis* 981 was collected from the coast of Xiapu (26°65′ N, 119°66′ E), Fujian Province, China. In the laboratory, the thalli were cultivated in the transparent aquaria at an irradiance of 30 µmol photons m^−2^·s^−1^ and a 12 h: 12 h light: dark cycle at 23 °C for one week. The tank contained 5 L filtered natural seawater (salinity of 25) filled with filtered air and the seawater was changed every day. These algal materials were used for correlation analysis, inhibition treatment and Western blotting analysis. 

### 4.2. Treatments for Correlation Analysis

The salinity and nitrogen experiments for the agar and enzyme correlation analysis were carried out in the laboratory. In each culture treatment, approximately 15 g of fresh thalli were placed into 10 L seawater and each treatment took three replicates. For the salinity treatment, the thalli were placed at a salinity of 17 and 33 with normal Provasoli medium, respectively. The samples for agar and enzymic analyses were collected on 0 day, 7 day, 14 day, 21 day, respectively. For the nitrogen treatment, the thalli were cultured in seawater (salinity of 25) with normal (HN) and nitrogen-limited Provasoli medium (LN), respectively. The samples for the agar and enzymic analyses were collected on 0 day, 7 day, 14 day and 21 day, respectively. Apart for the nitrogen and salinity factors, the other culture conditions were consistent with the adaptation culture condition.

In the cultivation months experiment, the alga of *G. lemaneiformis* 981 was collected from Xiapu in January, March, May, July, September, November of 2020. In the different cultivation fields’ experiment, the algae were collected from Xiapu, Lianjiang in Fujian Province, Xiangshan in Zhejiang Province in July of 2020, respectively. Intact and healthy thalli were selected from floating lines and placed in liquid nitrogen for analysis.

### 4.3. Agar Extraction and Enzymic Activities Assays

Approximately 5 g of fresh thalli from the aforementioned treatments were used to assay the agar content. The agar content of *G. lemaneiformis* was determined, using the alkali treatment method [6]. According to the manufacturer’s instructions, the agar content was calculated as the following formula: Agar content (%) = dry weight of agar / dry weight of thalli × 100.

The activities of the enzymes involved in the agar biosynthetic pathway were analyzed. Of them, PMI activity was detected according to Maruta et al. [37]; PGM and PMM activity was determined, according to Zhang et al. [18]. UGPase activity was assayed, using the ELISA kit (Kexing, Shanghai, China).

### 4.4. GlPGM Identification

The whole genome sequences of *G. lemaneiformis* were captured from our own laboratory database [38]. For catching the PGM protein sequence, the Hidden Markov Model (HMM) analysis was used for the search. The HMM profiles of the phosphoglucomutase/phosphomannomutase (PF02878, PF02879, PF02880 and PF00408) from the Pfam protein family database Pfam version 35.0 (http://pfam.xfam.org/, 29 June 2022) were used as the query to search the *G. lemaneiformis* protein database with an e-value ≤ e^−10^. Blastp was subsequently used to search for missing possible PGM candidates. Pfam and SMART (http://smart.embl-heidelberg.de/, 29 June 2022) were then used to confirm the conserved domain, and the protein sequences lacking the conserved domains were excluded. 

The total RNA from *G. lemaneiformis* was isolated by a RNeasy Plant mini kit (QIAGEN, Germany), according to the manufacturer’s protocol. The cDNA for the full-length sequence cloning was subsequently synthetized by using HiScript II Reverse Transcriptase Kit (Vazyme, Nanjing, China). The opening reading frames (ORFs) of three possible *GlPGM* were annotated as *GlPGM1*, *GlPGM2* and *GlPGM3*, and finally amplified by PCR, using Prime Star Kit (Takara, Dalian, China). The gene-specific primers with different restriction endonuclease site are indicated in the additional file Appendix A.

### 4.5. GlPGM Bioinformatic Analysis

The theoretical molecular weights and pI_s_ of GlPGM1, GlPGM2 and GlPGM3 were calculated by the ProtParam (https://web.expasy.org/protparam/, 29 June 2022). The conserved structural domains were constructed in the NCBI Conserved Domain Search. The protein sequences annotated as PGM from archaea, cyanobacteria, fungi, algae, and terrestrial plants were retrieved and collected from NCBI, JGI (https://genome.jgi.doe.gov/portal/, 29 June 2022), TAIR (https://www.arabidopsis.org/browse/genefamily/index.jsp, 29 June 2022) and the Ensembl (https://plants.ensembl.org/index.html, 29 June 2022) database, respectively. The PGMs of these species were presented on the additional file Appendix A. Multiple sequence alignment of PGM from different species was performed, using ClustalX, and MEME Suite (https://meme-suite.org/meme/, 29 June 2022) was used for searching the motifs of these PGM sequences. The phylogenetic tree was subsequently constructed, using maximum likelihood methods with 1000 bootstrap replicates. 

### 4.6. GlPGM Recombinant and Polyclonal Antibody Preparation

The ORFs of *GlPGM1* and *GlPGM3* were amplified, as mentioned above, and then cloned into the pMD18-T vector (Takara, Dalian, China) for sequencing (Youkang Biotech, Hangzhou, China). The *GlPGM1* and *GlPGM3* from the pMD18-T vector were then cut, using restriction endonuclease and cloned into expression vector pET-28a (+) (Takara, Dalian, China) to obtain pET-28a-*GlPGM1* and pET-28a-*GlPGM3* recombinant strains, respectively. The constructed plasmids were transformed into the expression strain *E. coli* BL21 (Takara, Dalian, China). The recombinant strains of GlPGM1and GlPGM3 were incubated and induced with 1mM isopropyl-β-D-thiogalactopyranoside (IPTG) for 3–5 h at 28 °C. The overexpressed proteins GlPGM1and GlPGM3 were purified and checked by 12.5% SDS-PAGE. The *E. coli* with recombinant pET-28a without IPTG was used as the negative control. 

The GlPGM1 and GlPGM3 polyclonal antibodies were made by Hangzhou Aiting Biological Technology Co., Ltd. (Hangzhou, China). The purified proteins, GlPGM1 and GlPGM3, were used as antigens to immunize rabbits for the production of the polyclonal antiserum. A Western blot analysis with the antibodies was conducted to check the expression of GlPGM1 and GlPGM3 from *G. lemaneiformis* (Appendix A).

### 4.7. Subcellular Localization

First, the online prediction tools were used to predict the possible subcellular localizations of GlPGM1, GlPGM2 and GlPGM3, using Cello v2.5 (http://cello.life.nctu.edu.tw/, 29 June 2022), WoLF PSORT tool (https://wolfpsort.hgc.jp/, 29 June 2022) and Plant-mPLoc (http://www.csbio.sjtu.edu.cn/bioinf/plant-multi/, 29 June 2022). 

Then, the immunoelectron microscopy was conducted to validate the subcellular localizations of GlPGM1 and GlPGM3. The thalli were fixed in 0.5% glutaraldehyde and 4% paraformaldehyde in 0.1 M phosphate buffer (pH 7.4) overnight and then dehydrated, using a gradient of ethanol concentrations in −20 °C. The dehydrated thalli were infiltrated with a gradient of LG Gold ethanol concentrations and embedded in 0.1% BENZIL in LG Gold ethanol for a week, followed by sectioning with a LEICA EM UC7 ultramicrotome instrument (Leica, Germany). The sections were blocked for 2 h at room temperature and then washed three times with TBST, followed by incubating with polyclonal antibody against GlPGM1 and GlPGM3, respectively. The sections were finally incubated with goat anti-rabbit IgG H&L (15 nm Gold) 1 h and examined with a TEM instrument (H-7650, Hitachi, Japan). 

### 4.8. Western Blot Analysis

Immunoblotting analysis was performed, as described in [39]. The total proteins were extracted from the different treatments by RIPA lysis buffer and quantified through bicinchoninic acid (BCA) protein assay kit (Beyotime, Shanghai, China). After adding 5 × SDS loading buffer in the samples and boiling for 10 min, the proteins were separated by 12.5% SDS-PAGE, using electrophoresis and then transferred to polyvinylidene difluoride membrane. After blocking, the membranes were then incubated overnight at 4 °C with a rabbit polyclonal antibody directed against GlPGM1 and GlPGM3 diluted 1/200 in the blocking buffer. Then the membranes were incubated 2 h at room temperature with horseradish peroxidase-labelled goat anti-rabbit, diluted 1/8000. The membranes were finally incubated 3–5 min in ECL (Advansta, CA, USA) and the signals were acquired by ChemiScope (Clinx, Shanghai, China) in dark condition. The second antibody, horseradish peroxidase-labelled goat anti-rabbit, was purchased from Abmart Co., Ltd. (Shanghai, China). 

### 4.9. Analysis of Enzyme Activities under PGM Inhibited Condition

The enzyme activities participated in agar biosynthetic pathway and other carbon metabolisms were analyzed at 0 h, 3 h, 12 h, 24 h, respectively. The activities of the enzymes (PGM, UGPase, PMI) were measured, using the similar methods described previously. The activities of possible (iso)floridoside synthase TPS and degrading enzyme GLA were detected, according to the previous descriptions [40,41], respectively. The activities of soluble starch synthase SS were detected, using the corresponding kit (Geruisi, Suzhou, China).

### 4.10. Carbon Metabolism Analysis under PGM Inhibited Condition

An Ultra Performance Liquid Chromatography-electrospray ionization-tandem mass spectrometry (UPLC-ESI-MS/MS) system was used to analyze the metabolites. The extraction, detection, qualification and quantification of the metabolites were performed at Metware Biotechnology Co., Ltd. (Wuhan, China), following the method described previously [42]. The floridean starch contents were assayed, according to the previous description [6]. The relative quantification of (iso)floridoside was performed by HPLC-MS described previously [43].

### 4.11. Statistical Analysis

The correlation between the enzyme activities and agar contents was conducted by Pearson correlation analysis (SPSS 16.0, USA). The normal distribution of all of the data under each treatment and the homogeneity of variance were confirmed by a Shapiro–Wilk test (*p* > 0.05) and Levene’s test (*p* > 0.05), respectively. The effects of the treatments, the time, and their interactions were assessed by a two-way analysis of variance (ANOVA). A Tukey post hoc test (Tukey HSD) was performed to show the differences between the treatments and time. The significance level was set at *p* < 0.05.

## 5. Conclusions

This study revealed the high correlation between PGM activities and agar contents in *G. lemaneiformis*, and explored the function of PGM in carbon metabolic pathways, especially in agar biosynthesis (Figure 7). GlPGM1 was located in the chloroplast, while GlPGM3 was revealed to have no specific localization. Western blot analysis confirmed that GlPGM1 and GlPGM3 in the different locations showed different functions in this seaweed, and GlPGM3 may have participated in (iso)floridoside, starch and agar synthesis. After the activities of PGM were limited by the inhibitor, Calvin cycle, and the respiratory metabolic pathways including EMP, TCA cycle and the agar biosynthesis pathway were downregulated, many of the lipids, phenolic acids and fructose-1,6-phosphate were significantly increased. Overall, despite the complexity of identifying the crucial metabolic nodes, our data reveal a possible carbon flux between the metabolites and agar, paving the way for tailoring the metabolic rewiring strategy for generating high agar-yielding thalli.

## Figures and Tables

**Figure 1 marinedrugs-20-00442-f001:**
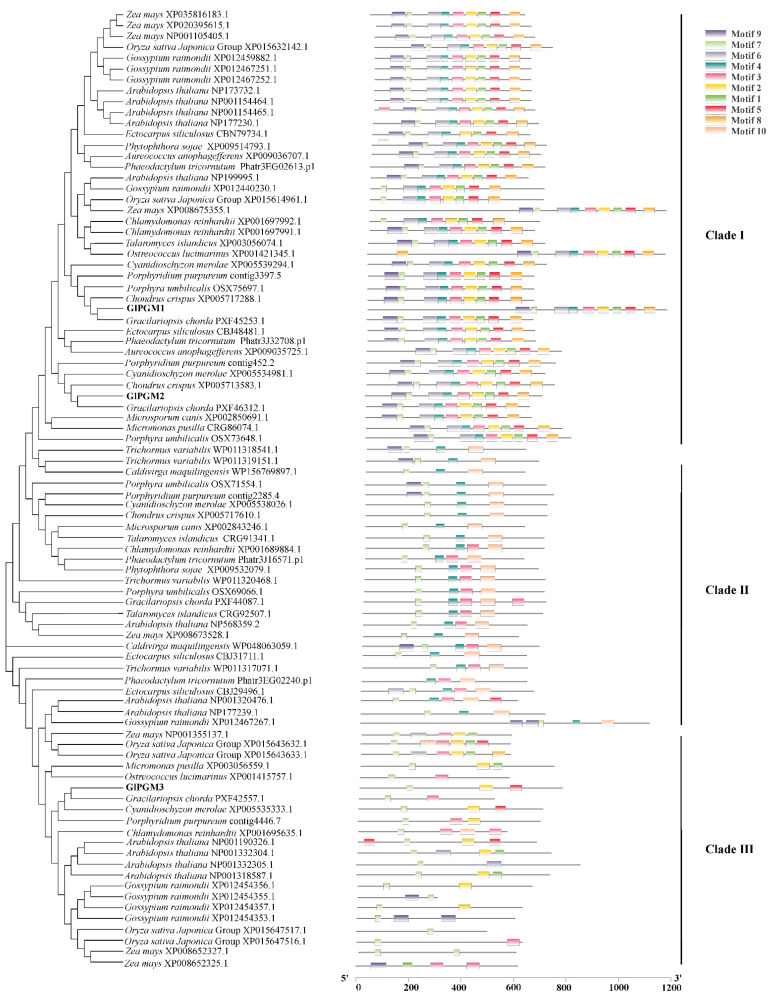
Phylogenetic tree and conserved motifs constructed based on 88 protein sequences using the maximum likelihood method. The bootstrap analysis was carried out with 1000 replicates.

**Figure 2 marinedrugs-20-00442-f002:**
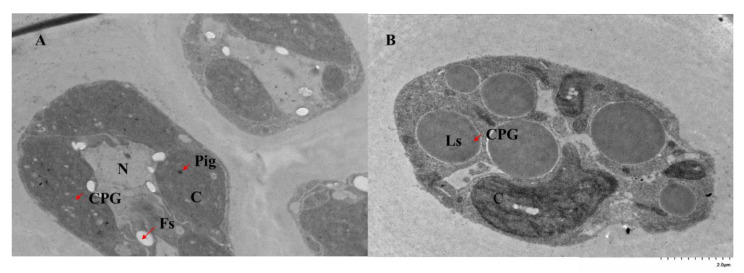
Subcellular location of GlPGM1 and GlPGM3 observed with transmission electron microscope. (**A**) the ultrathin section of *Gracilariopsis lemaneiformis* incubated with polyclonal antibody of GlPGM1 and identified by gold labeled secondary antibody (15 nm); (**B**) the ultrathin section of *Gracilariopsis lemaneiformis* incubated with polyclonal antibody of GlPGM3 and identified by gold labeled secondary antibody (15 nm). Abbreviations: C, the chloroplast in the cell; N, the nuclear of the cell; Fs, the floridean starch; Pig, the pigment; Ls, the liposome; CPG, colloidal gold particles.

**Figure 3 marinedrugs-20-00442-f003:**
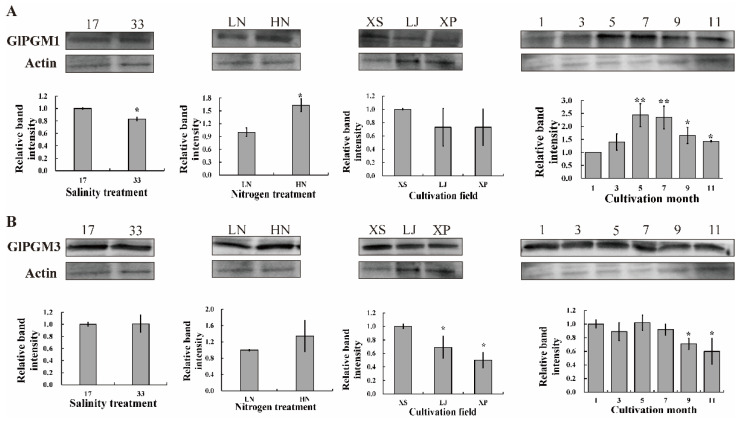
Western blot analysis of GlPGM1 (**A**) and GlPGM3 (**B**) in the different treatments. Abbreviations: 17 and 33 represent different salinity treatments; LN, treatment with nitrogen limited Provasoli medium; HN, treatment with normal nitrogen Provasoli medium; XS, LJ and XP represent different cultivation fields of Xiangshan, Lianjiang, Xiapu, respectively; 1, 3, 5, 7, 9 and 11, cultivation months in 2020 from Xiapu, Fujian, China. The intensity of *Gracilariopsis lemaneiformis* at first lane was set to 1, respectively. Actin was used as the control. * indicates significant difference between different treatments (* *p* < 0.05, ** *p* < 0.01).

**Figure 4 marinedrugs-20-00442-f004:**
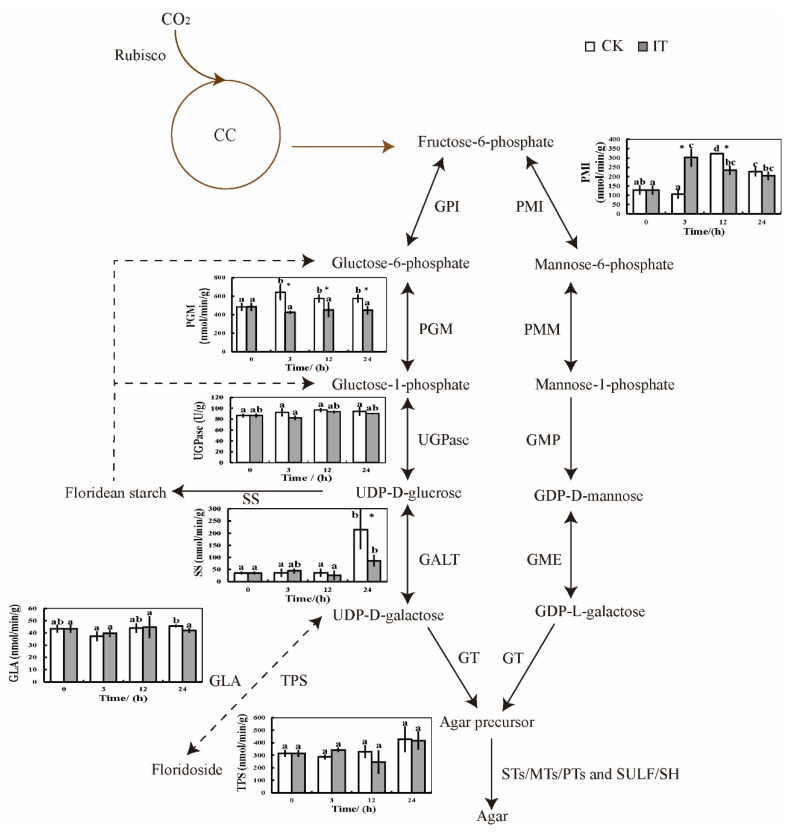
Changes of enzymic activities related to agar biosynthesis pathway in *Gracilariopsis lemaneiformis* during the 24 h period. In the column diagram, the different lowercase represents significant differences with experimental time. * indicates significant difference between different treatments (*p* < 0.05). Abbreviations: CK, control treatment; IT, PGM inhibitor treatment. CC, Calvin cycle; GPI, glucose-6-phosphate isomerase; PGM, phosphoglucomutase; UGPase, UDP-glucose pyrophosphorylase; GALT, galactose-1-phosphate uridylyltransferase; SS, floridean starch synthase; GLA, α-galactosidase; TPS, trehalose-6-phosphate synthase; PMI, mannose-6-phosphate isomerase; PMM, phosphomannomutase; GMP, GDP-mannose pyrophosphorylase; GME, GDP-mannose-3′,5′-epimerase; GT, galactosyltransferases; STs/MTs/PTs, sulfotransferase, methyltransferase and pyruvyltransferase; SULF/SH, sulfatase and sulfate hydrolase.

**Figure 5 marinedrugs-20-00442-f005:**
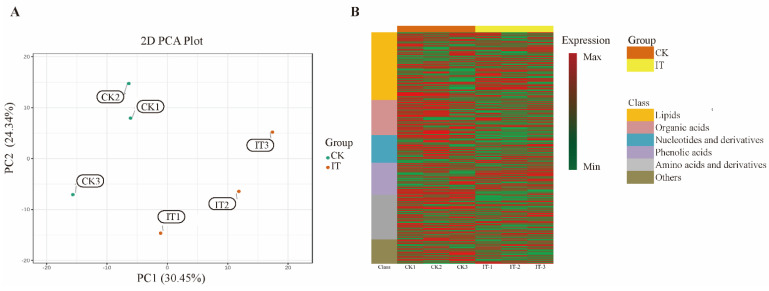
PCA (**A**) and heatmap (**B**) of primary carbon metabolites in *Gracilariopsis lemaneiformis*. Abbreviations: CK, control treatment; IT, PGM inhibitor treatment.

**Figure 6 marinedrugs-20-00442-f006:**
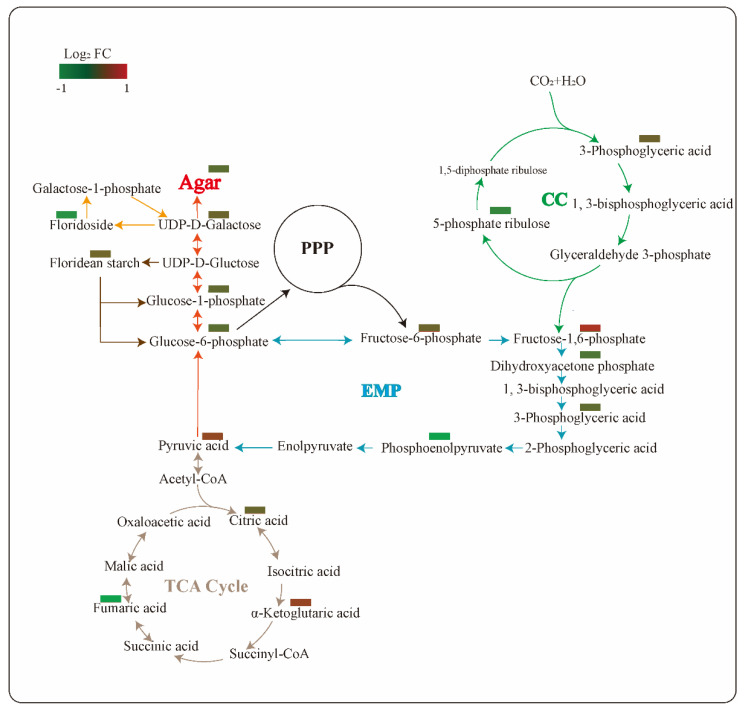
Changes of the identified metabolites in agar biosynthesis and carbon pathways in *Gracilariopsis lemaneiformis*. Abbreviations: FC, fold change of PGM inhibitor treatment (IT) compared to control treatment (CK); PPP, pentose phosphate pathway; EMP, Embden–Meyerhof–Parnas pathway; CC, Calvin cycle; TCA cycle, tricarboxylic acid cycle.

**Figure 7 marinedrugs-20-00442-f007:**
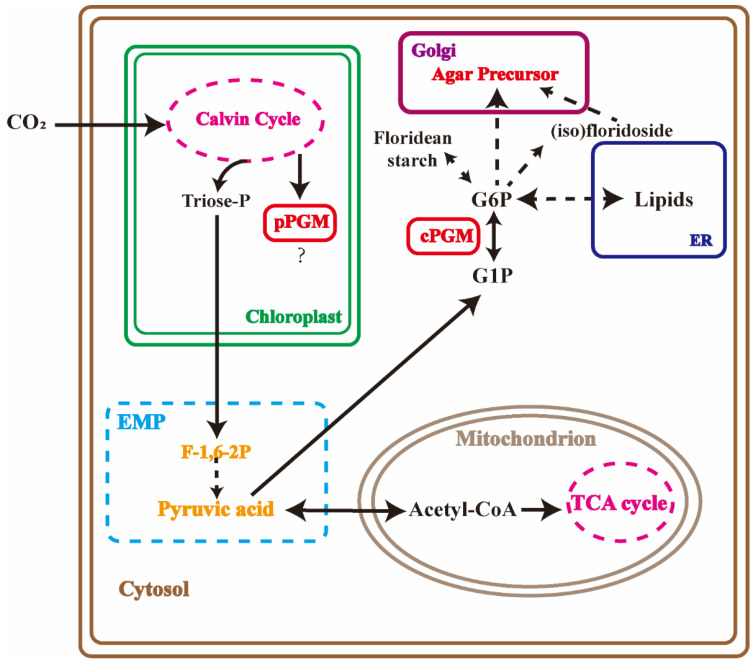
A possible carbon flux in *Gracilariopsis lemaneiformis*. EMP, Embden–Meyerhof–Parnas pathway; TCA cycle, tricarboxylic acid cycle; ER, endoplasmic reticulum; GAP, glyceraldehyde 3-phosphate; G6P, glucose 6-phosphate; G1P, glucose-1-phosphate; F-1, 6-2P, fructose-1, 6-bisphosphate.

**Table 1 marinedrugs-20-00442-t001:** Pearson correlation analysis between enzymic activities and agar contents in agar biosynthesis.

Variables		PMI	PGM	PMM	UGPase
Salinity treatment					
7 d	Agar	−0.095	0.184	−0.091	0.552
14 d	Agar	0.033	0.476	0.572	−0.017
21 d	Agar	−0.404	0.255	−0.779	0.213
Nitrogen treatment					
7 d	Agar	0.896 *	0.955 *	0.888 *	0.787
14 d	Agar	0.354	0.702	0.683	0.864 *
21 d	Agar	0.740	0.964 *	0.612	−0.980 *
Cultivation field	Agar	−0.043	0.999 *	0.460	0.332
Cultivation month	Agar	0.763	0.973 *	0.077	0.909

* indicates significant difference between enzymic activities and agar contents (*p* < 0.05).

## Data Availability

Not applicable.

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
