# Peer review of "Identification, Characteristics and Function of Phosphoglucomutase (PGM) in the Agar Biosynthesis and Carbon Flux in the Agarophyte Gracilariopsis lemaneiformis (Rhodophyta)"

_marinedrugs, 2022, doi:10.3390/md20070442_

Round 1
Reviewer 1 Report
The subject of the study (agar synthesis and the enzymes involved in this process and its relation to the climate conditions, and inhibitors) is very interesting with sufficient presentation and discussion. Only mention the missed three enzymes names UGPase, PMI, PMM in the abstract with PGM with their full name. In some places in the manuscript GIPGM1 and GIPGM2 and 3 genes were written with capital and regular style and in others they are written with small and italic style please uniform the style.
Author Response
We thank the editor and reviewer for the advices on the MS. We have completed all the modification according to the useful suggestions from the reviews.
Point 1: Only mention the missed three enzymes names UGPase, PMI, PMM in the abstract with PGM with their full name. In some places in the manuscript GIPGM1 and GIPGM2 and 3 genes were written with capital and regular style and in others they are written with small and italic style please uniform the style.
Response 1: According to the suggestions, we checked and edited all the enzyme and gene names. The GlPGM genes and other genes were all modified into capital and italic style form, and GlPGM proteins were still with capital and regular style form. For English language and style, we checked throughout the manuscript and made other minor modifications in the whole manuscript including the senteces, figures and suppelementary materials. And all these modifications were marked up using the track changes.

Reviewer 2 Report
In the manuscript by Qionglin Chen et al. “Identification, characteristic and function of phosphogluco- 2 mutase (PGM) in the agar biosynthesis and carbon flux in the agarophyte Gracilariopsis lemaneiformis (Rhodophyta)” are presented the results of a study of agar biosynthesis in commercial seaweed Gracilariopsis lemaneiformis. Among these agarophytes, Gracilariopsis lemaneiformis is rich in biologically active ingredients, has good agar yield and is mainly used for agar production. The experiments carried out with using various biological physico-chemical and computational methods are described in details. The activities of four enzymes involved in the agar biosynthesis was studied and it was confirmed that phosphoglucomutase (PGM) is closely related to agar accumulation. Three isoforms of PGM were characterized., their localization was established, their role in the metabolism of Gracilariopsis lemaneiformis studied. It was shown that these forms of this enzyme have different function and that one of them (GlPGM3) can be involved in the synthesis of (iso)floridoside, floridean starch and agar. The effect of PGM inhibition on the metabolic pathways has been studied and the essential role of PGM in regulating the carbon flux between agar and other carbohydrates has been confirmed. The article presents new data on the agar biosynthesis in Gracilariopsis lemaneiformis. However both introduction and discussion contain some redundant information that is not necessary for understanding the content of the article and is more suitable for a review. It is desirable to make the atricle shorter. The capture under the figure 3 does not contain the decoding of all abbreviation which were used. The article may be published after minor сorrections.
Author Response
We thank the editor and reviewer for the advices on the MS. We have completed all the modification according to the useful suggestions from the reviews.
Point 1: However both introduction and discussion contain some redundant information that is not necessary for understanding the content of the article and is more suitable for a review. It is desirable to make the atricle shorter.
Response 1: According to the suggestions, The introduction was cut into 120 lines, and discussion was reduced from 300 lines to 250 lines.
Point2: The capture under the figure 3 does not contain the decoding of all abbreviation which were used. The article may be published after minor сorrections.
Response 2: The captures of the figure 3 were supplemented according to the suggestions.
Apart from the above corrections, we have made other minor modifications in the whole manuscript including the senteces, figures and suppelementary materials. And all these modifications were marked up using the track changes.
